# Variation in Leaf Type, Canopy Architecture, and Light and Nitrogen Distribution Characteristics of Two Winter Wheat (*Triticum aestivum* L.) Varieties with High Nitrogen-Use Efficiency

**Zhiyong Zhang** [1,2,3], **Saijun Xu** [1,2,3], **Qiongru Wei** [1,2,3], **Yuxiu Yang** [1,2,3], **Huqiang Pan** [1,2,3], **Xinlu Fu** [1,2,3], **Zehua Fan** [1,2,3], **Butan Qin** [1,2,3], **Xiaochun Wang** [4], **Xinming Ma** [1,2,3] and **Shuping Xiong** [1,2,3,*]

1   College of Agronomy, Henan Agricultural University, Zhengzhou 450002, China
2   Collaborative Innovation Center of Henan Grain Crops, College of Agronomy, Henan Agricultural University, Zhengzhou 450000, China
3   Key Laboratory of Physiology, Ecology and Genetic Improvement of Food Crops in Henan Province, Zhengzhou 450002, China
4   College of Life Science, Henan Agriculture University, Zhengzhou 450002, China
*   Correspondence: shupxiong@henau.edu.cn

**Abstract:** Studies of traits related to nitrogen (N)-use efficiency (NUE) in wheat cultivars are important for breeding N-efficient cultivars. Canopy structure has a major effect on NUE, as it determines the distribution of light and N. However, the mechanism by which canopy structure affects the distribution of light and N within the canopy remains unclear. The N-efficient winter wheat varieties YM49 and ZM27 and N-inefficient winter wheat varieties XN509 and AK58 were grown in the field under two N levels. Light transmittance was enhanced, and the leaf area index and photosynthetically active radiation were lower in the N-efficient cultivar population, which was characterized by moderately sized flag leaves, a low frequency of canopy leaf curling, a low light attenuation coefficient ($K_L$), and high plant compactness. Reductions in the amount of shade increased the distribution of light and N resources to the middle and lower layers. The photosynthetic rate, transpiration rate, instant water-use efficiency, and canopy photosynthetic NUE were higher, N remobilization of the upper and middle canopy leaves was reduced, and the leaf N content was high in the N-efficient cultivars. A higher ratio of the N extinction coefficient ($K_N$) to $K_L$ reflects the assimilation ability of the N-efficient winter wheat cultivars, resulting in improved canopy structure and distribution of light and N, higher 1000-grain weight and grain yield, and significantly increased light and NUE. An improved match between gradients of light and N in the leaf canopy promotes balanced C and N metabolism and reduces energy and nutrient losses. This should be a goal when breeding N-efficient wheat cultivars and implementing tillage regimes.

**Keywords:** nitrogen-use efficiency; canopy architecture; leaf type; light gradient; nitrogen gradient; winter wheat (*Triticum aestivum* L.)

## 1. Introduction

Wheat (*Triticum aestivum* L.) is the most widely grown crop, and it provides approximately 20% of the calories and protein consumed worldwide [1,2]. Crop yield is greatly limited by nitrogen (N) levels. Large amounts of N fertilizer, including synthetic N fertilizer, are often applied to enhance wheat productivity in China [3]. However, N fertilizer addition can have negative effects on crop yield when excessive amounts of N are applied [4]. Indeed, more than 70% of the N in agricultural fields in China is not effectively utilized [5]. The N remaining after it is utilized by crops to support their growth is rapidly lost to the environment through ammonia volatilization, denitrification, surface runoff, and other pathways [6]. This results in soil consolidation, reductions in crop yield and quality [7], eutrophication and acidification of aquatic and terrestrial ecosystems [8], and

the formation of nitrous oxide, which contributes to the greenhouse effect [9,10]. These findings explain why the N-use efficiency (NUE) of agricultural fields in China is only 30–40%, which is lower than the global average NUE and more than 10% lower than that in North America or Europe, where it is typically above 50% [11]. Considering current NUE estimates, three times the amount of N fertilizer currently applied will be needed by 2050 to meet a projected increase in global food demand of 70% and feed a projected global population of 9.7 billion people [4,10]. For this reason, enhancing the NUE of winter wheat is important for achieving food security and preventing environmental degradation both in China specifically, as well as in the world more generally. Characterizing genetic differences in N metabolism in winter wheat is critically important for the breeding of N-efficient cultivars with higher NUE [12–14].

NUE is defined as the total biomass or grain yield produced per unit of applied N fertilizer [15]. It has two primary components: N uptake efficiency (plant N uptake/soil available N, NUpE) and N utilization efficiency (grain yield/total N uptake, NUtE). NUtE indicates the translocation and remobilization capacity of N in plants, which is affected by the canopy light (photosynthetically active radiation, PAR) distribution and canopy interception capacity; the PAR distribution and canopy interception capacity are determined by the structural features of canopy leaves and plants [14,16]. Light and gas conditions of interplant canopies, as well as the light transmittance and $CO_2$ fixation ability of crops, are improved when they are dominated by narrow, erect leaves and compact plants, especially in the middle and lower layers [16–18]. Erect leaf angle is thought to increase tolerance of high planting densities [19] by significantly enhancing the PAR conversion efficiency (PCE) and PAR use efficiency (PUE) of wheat [16,20,21].

The canopy N and light distribution can be characterized by the N extinction coefficient ($K_N$) and light extinction coefficient ($K_L$), respectively [22]. Effective canopies facilitate access to light energy by deep leaves, enhance the photosynthetic rate of leaves [23], and provide material and energy to support the accumulation of photosynthetic N. A previous study indicates that more than 75% of the total N content in leaves might be involved in photosynthesis [24]. Canopy photosynthesis is increased by at least 20% when gradients of N and light are consistent (i.e., $K_N = K_L$) [25,26]. Previous studies examining the effects of leaf type have mainly focused on flag leaves; by contrast, few studies have examined the vertical distribution of leaf types in the canopy. In $C_3$ crops, the light saturation point of flag leaf photosynthesis is achieved when the solar-to-product energy conversion efficiency is a quarter of the total incident sunlight energy [27,28]. Canopy photosynthesis still shows a high potential to significantly contribute to yield improvements [29,30]. Canopy N, the light distribution, and canopy photosynthetic N-use efficiency (PNUE) can be substantially enhanced under different management regimes by modifying the distribution of light [22,31,32]. Previous studies have mainly focused on comparing canopy N, the light distribution, and PNUE among different varieties [33–37]; the relationships between plant type, canopy light, and the N distribution during wheat production have been less explored by comparison. Here, we determined the phenotype, canopy structure, and N and light distribution characteristics of N-efficient winter wheat cultivars from populations of two types of NUE winter wheat varieties under different N application levels. Overall, the results of our study provide new insights that will aid the breeding and cultivation of N-efficient winter wheat varieties.

## 2. Materials and Methods

### 2.1. Plant Materials, Field Management Practices, and Growth Conditions

The experiment was conducted at the experimental farm of Henan Agricultural University (Xuchang, Henan, China, 34°13′3.56″ N, 113°80′3.84″ E) from 2020 to 2021. Monthly accumulated precipitation, monthly accumulated total solar radiation, and monthly mean temperature during the winter wheat growing season (October to June) for 2020–2021 are shown in Figure 1. Soybean was grown at the experimental farm before our experiment; the chemical properties of the loamy soil before N treatment are shown in Table 1. The ex-

periment was conducted in a split–split plot design with two factors and three replications. There were two N levels in the main plots: N8 (120 kg·hm$^{-2}$) and N15 (225 kg·hm$^{-2}$). Four cultivars were planted in the subplots: two N-inefficient cultivars (XN509 and AK58) and two N-efficient cultivars (YM49 and ZM27). N fertilizer was applied in the form of urea, and the N content was 46%. Two-thirds of the N fertilizer was applied as a starter fertilizer, and one-third of the N fertilizer was applied as a topdressing at the jointing stage. Phosphate and potassium fertilizers (16% superphosphate (P$_2$O$_5$) and 60% potassium chloride (K$_2$O), respectively) were applied before sowing at 105 kg·ha$^{-1}$. All other agricultural practices were the same as those used in winter wheat fields managed for yield maximization.

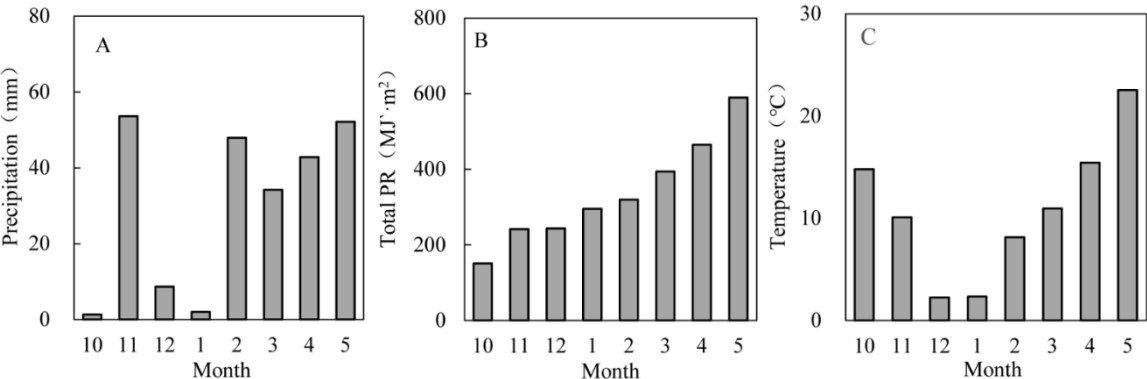

**Figure 1.** Monthly accumulated rainfall ((**A**), mm), monthly accumulated total solar radiation ((**B**), MJ·m$^{-2}$), and monthly mean temperature ((**C**), °C) during the winter wheat growing season (October to June) in 2020–2021.

**Table 1.** Chemical properties of the experimental field soil before N treatment.

| Year | Organic Matter | Total N | Alkaline N | Available P | Available K |
|---|---|---|---|---|---|
| | (g·kg$^{-1}$) | (g·kg$^{-1}$) | (mg·kg$^{-1}$) | (mg·kg$^{-1}$) | (mg·kg$^{-1}$) |
| 2020–2021 | 16.53 | 0.64 | 27.99 | 10.07 | 352.43 |

*2.2. Traits and Measurements*

2.2.1. Leaf Morphological Traits

Twenty uniform flag leaves of wheat during later reproductive growth stages were randomly selected for measurements. The flag leaf length (FLL, from the base of the ligula to the tip of the leaf, cm), flag leaf width (FLW, widest part of the leaf, cm), flag leaf area (FLA, area of the entire leaf, cm$^2$), and flag leaf perimeter (FLP, perimeter of the entire leaf, cm) were measured using a CI-203 Portable Area Meter (CID Bio-Science, Inc., Vancouver, WA, USA). Flag leaf natural width (FLNW, widest part of the leaf when the leaf is not dehydrated, cm) was measured using a ruler, and leaf curvature (LC) was measured using the following formula: LC = FLW/FLNW.

The wheat canopy was separated into three layers (Figure 2): the upper layer (UL) was above the base of the flag leaves, the middle layer (ML) was between the base of the flag leaves and inverted second leaves, and the lower layer (LL) was between the base of the inverted second leaves and 20 cm above the ground. The leaf base angle (BA) and leaf open angle (OA) of the three layers were measured using a protractor and the leaf drooping angle (DA) = BA − OA.

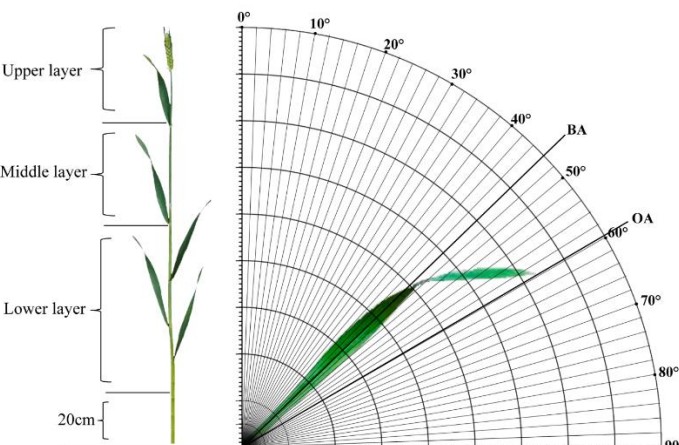

**Figure 2.** Canopy layer and leaf angle. BA: base angle. OA: open angle.

### 2.2.2. Leaf Area Index

The leaf area index (LAI) of the three layers of the wheat canopy was measured at the base of each layer and between 10:00 and 14:00 at 7 d after flowering at three random points in each plot using a plant canopy analyzer (LAI 2200C, LI-COR, Lincoln, NE, USA). Measurements were taken three times at each point, and the average values were used in subsequent analyses.

### 2.2.3. Biomass

Ten uniform plants per plot were separated into green leaves, culm, and panicle at harvest for experiments at 7 d after flowering and the maturity period. The harvested plants were dried initially at 105 °C for 1 h to ensure that enzymes were deactivated; they were then oven-dried at 80 °C to constant weight before the weight was taken.

### 2.2.4. PAR

PAR values of the wheat canopy were determined at the top of the canopy and at the base of each layer between 14:00 and 16:00 at 7 d after flowering at three random points in each plot using an AccuPAR canopy analyzer (LP-80, Decagon Devices Inc., Pullman, WA, USA). Measurements were taken three times at each point, and the average values were used in subsequent analyses.

### 2.2.5. CaR, IPAR, PCE, and PUE

Light interception and light-energy utilization can be described by the following exponential functions:

$$CaR = (PAR_n - PAR_{n-1})/PAR_t \times 100\% \tag{1}$$

$$IPAR = S \times CaR_t \times 0.5 \tag{2}$$

$$PCE = NDMM/IPAR \tag{3}$$

$$PUE = CaR_t \times PCE \tag{4}$$

where $PAR_t$ corresponds to PAR at 50 cm above the top of the canopy, n indicates the canopy layer, and S stands for the amount of solar radiation from the experimental farm from the fraction of radiation intercepted ($CAR_t$) by the plant canopy and the intercepted PAR (IPAR). The net accumulation of dry matter (NDMM, g·m$^{-2}$) was calculated and integrated across 7 d after flowering and the maturity period. PCE (g·MJ$^{-1}$) corresponds to PAR conversion efficiency, and PUE (g·MJ$^{-1}$) indicates PAR-use efficiency.

### 2.2.6. Photosynthetic Parameters

The photosynthetic rate ($P_n$) and transpiration rate ($T_r$) of the three layers of the wheat canopy were determined at the base of each layer between 10:00 and 14:00 at 7 d after flowering. A Li-6400 Portable Photosynthesis System (Li-COR, Lincoln, NE, USA) was used in an open 'sun and sky' chamber to measure three random points in each plot. Measurements were taken three times at each point, and average values were used in subsequent analyses.

### 2.2.7. Instant Water-Use Efficiency ($WUE_{inst}$)

The leaf water status relationship can be described by the following exponential function:

$$WUE_{inst} = P_n/T_r \tag{5}$$

### 2.2.8. $K_L$

The relationship between the canopy light distribution and canopy structure can be described by the following exponential function [38]:

$$I = I_0 \times \exp(-K_L \times F) \tag{6}$$

where F is the cumulative LAI from the top of the canopy; I and $I_0$ are the high photosynthetic photon flux density (PPFD) at F and the top of the canopy, respectively; and $K_L$ is the light extinction coefficient, which indicates the distribution of canopy light. A lower $K_L$ corresponds with a more compact canopy architecture and more uniform light distribution.

### 2.2.9. Yield and Yield Components

Spike number per $hm^2$, total grain number per spike, and 1000-grain weight were determined in two one-meter row sections by two random points in each plot for yield calculations.

### 2.2.10. N Content

The dried samples from above (see Biomass) were pulverized and digested at 380 °C using $H_2SO_4$-$H_2O_2$ [39], and the N content was determined using a Type AA3 continuous flow analyzer (AutoAnalyzer-3, SEAL Analytical, Inc., Norderstedt, Germany).

### 2.2.11. $K_N$

The distribution and profile of N within a canopy can be described using the following model [22]:

$$SLN_i = (N_0 - N_b) \times \exp(-K_N \times F) + N_b \tag{7}$$

where $SLN_i$ is the leaf N (g N $m^{-2}$) of the layer of the canopy, $N_0$ is the SLN of the top leaves, $K_N$ is the extinction coefficient for effective leaf N, and $N_b$ is the base value of leaf N for photosynthesis, which corresponds to the non-photosynthetic N content. Regardless of the canopy N content, canopy photosynthesis is maximized when $K_N = K_L$.

### 2.2.12. NUE

N accumulation in each organ is the product of the N content and biomass. N utilization efficiency (NUtE) is the grain yield divided by the amount of N accumulated in the plant at maturity [15].

### 2.2.13. Canopy Photosynthetic NUE (PNUE)

PNUE is defined as the ratio of gross canopy photosynthesis to the canopy leaf N content [1].

*2.3. Data Processing and Analysis*

Analyses of variance followed by post hoc tests were conducted using IBM SPSS Statistics (Version of 25). Microsoft Office Excel 2016 was used to process the data. Adobe Photoshop 2017 and Adobe Illustrator 2017 were used to make figures.

**3. Results**

*3.1. Yield, Yield Components, and NUtE*

The NUtE, NpUE, 1000-grain weight, and yield were 9.9–20.5%, 5.4%, 14.4–34.6%, and 18.5–30.1% higher in the N-efficient cultivars YM49 and ZM27 than in the N-inefficient cultivars XN509 and AK58, respectively, and all these differences were significant (Tables 2 and S1). There were no significant differences in the spike number and grain number per spike between N-efficient and N-inefficient cultivars. The NUtE, spike number, grain number per spike, and yield significantly increased and the 1000-grain weight significantly decreased as the amount of N applied increased.

**Table 2.** Yield, yield components, and NUtE of two N-efficient and two N-inefficient winter wheat varieties under different N levels.

| | SN ($\times 10^4 \cdot hm^2$) | GNS | 1000-GW (g) | GY (kg·hm$^{-2}$) | NUtE (g·g$^{-1}$) | NUpE (g·g$^{-1}$) |
|---|---|---|---|---|---|---|
| **Cultivar** | 27.66 ** | 335.02 ** | 1432.22 ** | 1242.41 ** | 4249.11 ** | 157.96 ** |
| XN509 (TN-in) | 511.87c | 41.87a | 34.84c | 7461b | 25.75c | 0.93c |
| AK58 (TN-in) | 523.29b | 36.44c | 40.98b | 7812b | 27.47b | 0.91d |
| YM49 (TN) | 537.72a | 36.72c | 46.88a | 9255a | 30.19a | 0.98b |
| ZM27 (TN) | 521.38b | 39.65b | 46.92a | 9705a | 31.02a | 1.00a |
| **N level** | 0.12 ** | 58.55 ** | 10.92 ** | 1654.88 ** | 632.53 ** | 1961.73 ** |
| N8 | 489.86b | 38.13b | 42.66a | 7930b | 29.08a | 1.03a |
| N15 | 557.26a | 39.21a | 42.16b | 9186a | 28.14b | 0.88b |
| **N level × Cultivar** | 60.64 ** | 47.23 ** | 1.94NS | 50.51 ** | 379.61 ** | 11.83 ** |

TN-in: The N-inefficient cultivars, TN: The N-efficient cultivars, SN: Spike number, GNS: Grain number, 1000-GW: 1000-grain weight, GY: Grain yield. Results of split–split plot variance analyses. NS, and ** indicate non-significant, significant at $p \leq 0.05$, and significant at 0.01, respectively. Different letters within each column indicate significant differences according to Duncan's multiple-range test ($p = 0.05$). The main effects were compared using Student's *t*-test.

*3.2. Leaf Morphological Traits, LAI, and Biomass*

3.2.1. Flag Leaf Morphological Traits

Flag leaf is the main vegetative organ of wheat for photosynthesis, and cultivars with different nitrogen efficiency types show different characteristics of flag leaf morphology at two nitrogen levels. Overall, the FLL, FLW, FLA, FLP, FLNW, and LC indexes of nitrogen-efficient varieties and nitrogen-inefficient varieties differed significantly between different varieties (Tables 3 and S2, Figure 3). Among them, FLL, FLW, FLA, FLP, and FLNW are irregular among different nitrogen efficiency varieties, but LC is manifested as nitrogen-efficient varieties higher than nitrogen-inefficient varieties, and nitrogen-efficient varieties are 7–15.8% higher than nitrogen-inefficient varieties. The LC of the high-efficiency nitrogen variety was between 1.2 and 1.3, and the LC of the nitrogen-inefficient variety was between 1.0 and 1.2. With the increase in nitrogen application amount, the FLL, FLW, FLAP, FLP, and FLNW of high-efficiency nitrogen and nitrogen-inefficient varieties increased significantly. However, LC did not vary with the amount of N administered. The flag leaf morphological trait of AK58 had the highest value among all varieties and the lowest value of XN509.

**Table 3.** Flag leaf morphological traits of two N-efficient and two N-inefficient winter wheat varieties under different N levels.

| | FLL | FLW | FLA | FLP | FLNW | LC |
|---|---|---|---|---|---|---|
| | (cm) | (cm) | (cm²) | (cm) | (cm) | |
| **Cultivar** | 840.55 ** | 792.36 ** | 728.42 ** | 867.95 ** | 4496.2 ** | 388.36 ** |
| XN509 (TN-in) | 15.45d | 1.63d | 18.91d | 30.15d | 1.51c | 1.08d |
| AK58 (TN-in) | 19.10a | 2.00a | 28.58a | 37.39a | 1.76a | 1.14c |
| YM49 (TN) | 18.95b | 1.94b | 27.66b | 37.19b | 1.55b | 1.25a |
| ZM27 (TN) | 17.71c | 1.81c | 24.23c | 34.50c | 1.49d | 1.22b |
| **N level** | 0.41 ** | 0.02 ** | 2.50 ** | 3.31 ** | 0.22 ** | 0.40NS |
| N8 | 17.67b | 1.84b | 24.52b | 34.44b | 1.57b | 1.17a |
| N15 | 17.93a | 1.85a | 25.16a | 35.18a | 1.58a | 1.17a |
| **N Level × Cultivar** | 5.78NS | 3.12NS | 5.15NS | 8.53 ** | 206.60 ** | NS |

TN-in: The N-inefficient cultivars, TN: The N-efficient cultivars. Results of split–split plot variance analyses. NS, and ** indicate non-significant, significant at $p \leq 0.05$, and significant at 0.01, respectively. Different letters within each column indicate significant differences according to Duncan's multiple-range test ($p = 0.05$). The main effects were compared using Student's *t*-test.

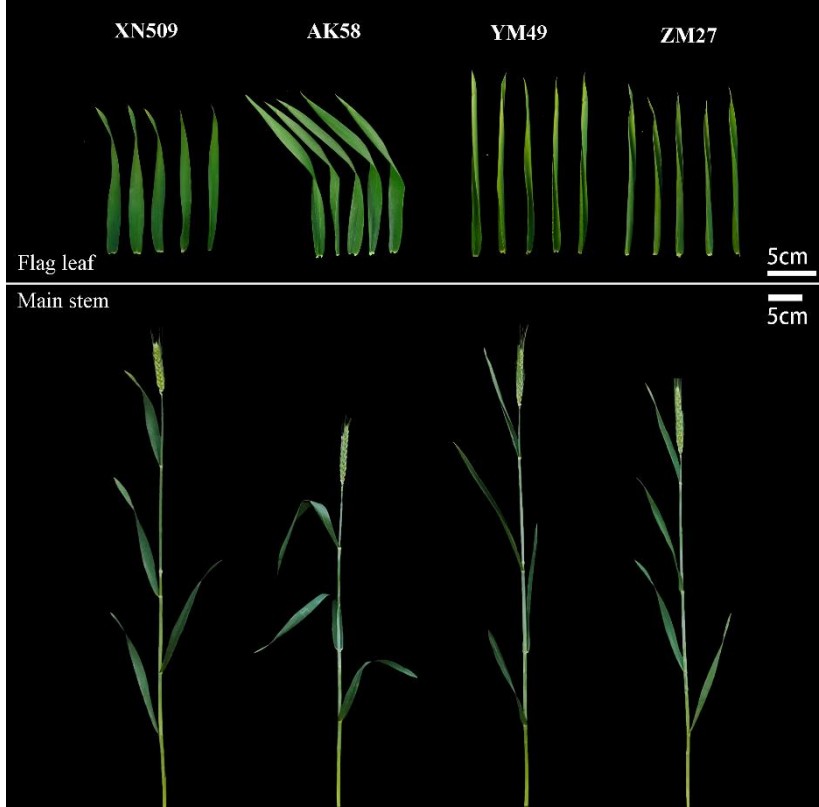

**Figure 3.** Leaf morphological traits of two N-efficient and N-inefficient winter wheat varieties under 225 kg·hm$^{-2}$ N.

### 3.2.2. Vertical Distribution of Leaf Angle

There were significant differences in BA, OA, and DA between N-efficient and N-inefficient cultivars (Tables 4 and S3, Figure 3). The BA was significantly higher and lower in N-efficient cultivars than in XN509 and AK58, respectively; in addition, the OA and DA were lower in the N-efficient cultivars than in the N-inefficient cultivars. Specifically, the BA was 7.8–21.8%, 9.9–19.1%, and 15.4–53.5% higher in AK58 than in YM49, ZM27, and XN509, respectively. The OA was 2.3–2.9 times, 2.2–3.1 times, and 1.8–3.1 times higher in AK58 than in YM49, ZM27, and XN509, respectively. The DA of AK58 was 11.9–25.6 times, 13.9–19.5 times, and 5.6–6.7 times higher in AK58 than in YM49, ZM27,

and XN509, respectively. The BA, OA, and DA of the N-efficient and N-inefficient cultivars were highest in the lower layer, followed by the middle layer and the upper layer, and differences among layers were significant. The BA, OA, and DA significantly increased as the amount of N applied increased.

**Table 4.** The leaf angle vertical distribution in two N-efficient and two N-inefficient winter wheat varieties under different levels of N.

| | BA (°) | | | OA (°) | | | DA (°) | | |
|---|---|---|---|---|---|---|---|---|---|
| | UL | ML | LL | UL | ML | LL | UL | ML | LL |
| **Cultivar** | 2643.47 ** | 1050.35 ** | 231.94 ** | 25,246.61 ** | 26,853.54 ** | 17,912.86 ** | 32,986.57 ** | 48,367.56 ** | 34,983.04 ** |
| XN509 (TN-in) | 13.90d | 18.00d | 20.31d | 21.65b | 25.69b | 29.46b | 7.73b | 7.66b | 9.15b |
| AK58 (TN-in) | 21.34a | 23.84a | 23.44a | 80.56a | 83.53a | 83.47a | 59.25a | 59.70a | 60.03a |
| YM49 (TN) | 17.52c | 18.99c | 21.75b | 19.74d | 22.82d | 26.42c | 2.23d | 3.83c | 4.67c |
| ZM27 (TN) | 17.92b | 20.32b | 21.32c | 20.81c | 24.01c | 25.33d | 2.89c | 3.69c | 4.02d |
| **N level** | 342.72 ** | 599.91 ** | 485.45 ** | 8160.33 ** | 9708.54 ** | 7090.5 ** | 10,990.31 ** | 16,383.67 ** | 11,992.5 ** |
| N8 | 17.13b | 19.32b | 20.76b | 27.19b | 30.08b | 32.28b | 10.06b | 10.76b | 11.52b |
| N15 | 18.22a | 21.25a | 22.65a | 44.21a | 47.94a | 50.01a | 25.99a | 26.91a | 27.62a |
| **N Level × Cultivar** | 59.05 ** | 35.51 ** | 23.46 ** | 7017.65 ** | 6453.54 ** | 5063.78 ** | 10,224.24 ** | 14,507.5 ** | 11,243.13 ** |

TN-in: The N-inefficient cultivars, TN: The N-efficient cultivars. Results of split–split plot variance analyses. ** indicate significant at $p \leq 0.05$. Different letters within each column indicate significant differences according to Duncan's multiple-range test ($p = 0.05$). The main effects were compared using Student's *t*-test.

### 3.2.3. Vertical Distribution of LAI

The LAI of the N-efficient cultivars was 4.9–21.8% and 16.7–35.6% lower in the upper and lower layers compared with that of the N-inefficient cultivars, respectively, and the LAI of the N-efficient cultivars was 39.6–87.5% higher compared with that of the N-inefficient cultivars in the middle layer; all these differences were significant (Figure 4). The LAI of the N-efficient and N-inefficient cultivars was highest in the lower layer, followed by the upper layer and the middle layer, and differences among layers were significant. The canopy LAI of all cultivars significantly increased as the amount of N applied increased.

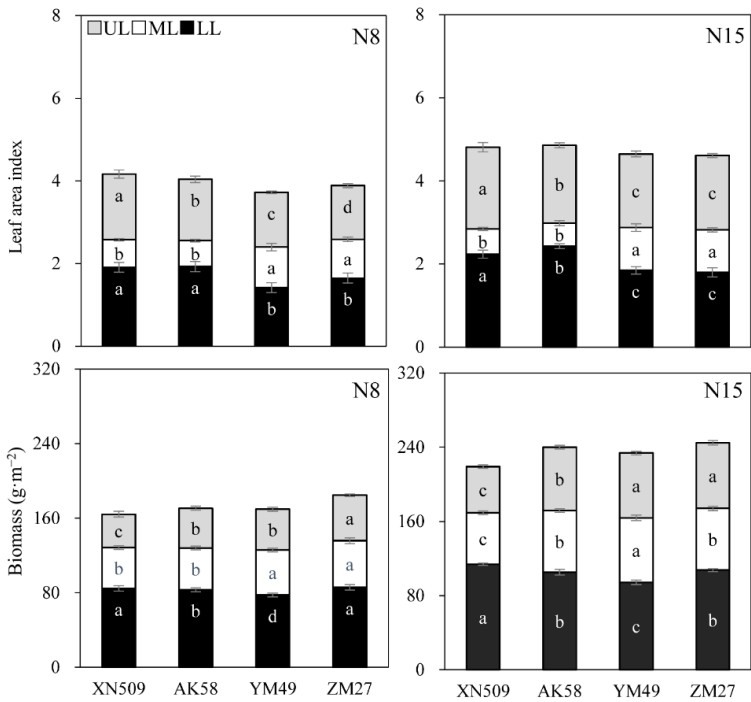

**Figure 4.** The vertical distribution of LAI and biomass of two N-efficient and two N-inefficient winter wheat varieties under different N levels. UL: upper layer. ML: middle layer. LL: lower layer. Different letters within each column indicate significant differences according to Duncan's multiple-range test ($p = 0.05$).

### 3.2.4. Vertical Distribution of Biomass

There were no significant differences in biomass between the N-efficient and N-inefficient cultivars in the upper and lower layers (Figure 4). The biomass of the N-efficient cultivars was 7.7–13.9% higher than that of the N-inefficient cultivars in the N8 treatment, and these differences were significant. The biomass of the N-efficient cultivars was only 19.8–25.2% higher than that of XN509 in the N15 treatment (these differences were significant), and there were no significant differences in the biomass of the N-efficient cultivars and AK58. The biomass of the N-efficient and N-inefficient cultivars was highest in the lower layer, followed by the middle layer and the upper layer, and the differences among layers were significant. The canopy biomass of the N-efficient and N-inefficient cultivars significantly increased as the amount of N applied increased.

### 3.3. Vertical Distribution of PAR, PCE, PUE, and Photosynthetic Parameters

#### 3.3.1. Vertical Distribution of PAR

The PAR of the upper layer was 14.9–30.4% lower in the N-efficient cultivars than in the N-inefficient cultivars. The PAR of the middle layer was 4–34% and 28.7–36.6% higher in the N-efficient cultivars than in the N-inefficient cultivars in the N8 and N15 treatments, respectively (Figure 5). The PAR of the lower layer was 13.8–96.9% higher in the N-efficient cultivars than in the N-inefficient cultivars. The PAR of the N-efficient and N-inefficient cultivars was highest in the lower layer, followed by the middle layer and the upper layer, and differences among layers were significant. The PAR of the N-efficient and N-inefficient cultivars significantly increased in the upper and middle layers as the amount of N applied increased; the PAR of the N-efficient and N-inefficient cultivars significantly decreased in the upper and middle layers as the amount of N applied increased.

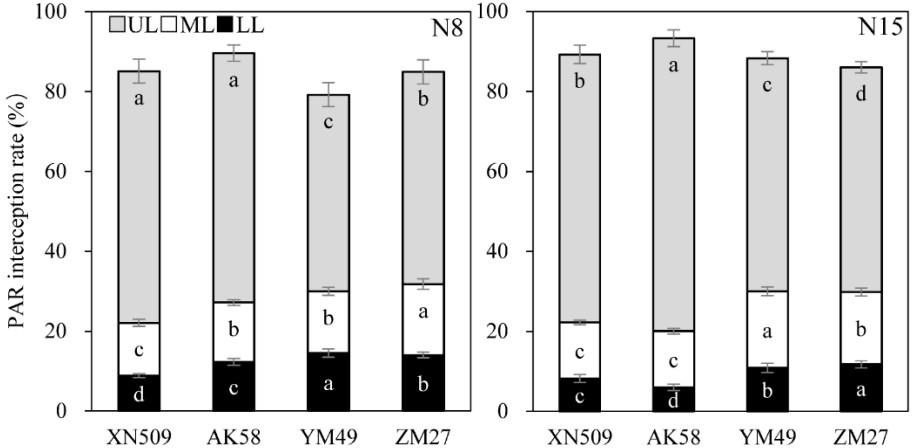

**Figure 5.** The vertical distribution of PAR of two N-efficient and two N-inefficient winter wheat varieties under different N levels. UL: upper layer. ML: middle layer. LL: lower layer. Different letters within each column indicate significant differences according to Duncan's multiple-range test ($p = 0.05$).

#### 3.3.2. DMW, PCE, and PUE

The DMW, PCE, and PUE were 12.4–29.3%, 38.7–83.1%, and 22.1–81.2% higher in the N-efficient cultivars than in the N-inefficient cultivars at the filling and maturity stages, respectively (Table 5). The IPAR values of the N-efficient cultivars were 1–13.5% lower in the N-efficient cultivars than in the N-inefficient cultivars, and these differences were significant. The DMW, IPAR, PCE, and PUE of the N-efficient and N-inefficient cultivars increased as the amount of N applied increased.

**Table 5.** The vertical distribution of DMW, PCE, and PUE of two N-efficient and two N-inefficient winter wheat varieties under different N levels.

| N Level | Cultivar | DMW (g·m$^{-2}$) | | IPAR | PCE | PUE |
|---|---|---|---|---|---|---|
| | | Filling Stage | Maturity Stage | (MJ·m$^{-2}$) | (g·MJ$^{-1}$) | (g·MJ$^{-1}$) |
| N8 | XN509 | 1595.99d | 1480.75c | 159.91b | 0.72c | 0.59d |
| | AK58 | 1632.37c | 1486.73d | 168.62a | 0.86b | 0.75c |
| | YM49 | 1849.64b | 1671.58b | 148.59c | 1.20a | 0.92b |
| | ZM27 | 1890.33a | 1689.94a | 159.54b | 1.26a | 1.03a |
| N15 | XN509 | 1809.78d | 1674.21d | 167.83b | 0.81c | 0.70d |
| | AK58 | 1887.09c | 1716.23c | 175.62a | 0.97b | 0.88c |
| | YM49 | 2340.83a | 2095.20a | 166.10d | 1.48a | 1.27a |
| | ZM27 | 2219.12b | 1995.83b | 161.68c | 1.38a | 1.15b |

Different letters within each column indicate significant differences according to Duncan's multiple-range test (*p* = 0.05).

### 3.3.3. Vertical Distribution of $P_n$, $T_r$, and $WUE_{inst}$

The $P_n$, $T_r$, and $WUE_{inst}$ of the N-efficient cultivars were 41.2–86.8%, 6.7–44.7%, and 17.8–45.2% higher in the N-efficient cultivars than in the N-inefficient cultivars in the upper and middle layers, respectively, and these differences were significant (Table 6). $P_n$ and $T_r$ were significantly higher in the N-efficient cultivars than in the N-inefficient cultivars in the lower layer, and the opposite pattern was observed for $WUE_{inst}$. The $P_n$, $T_r$, and $WUE_{inst}$ of the N-efficient and N-inefficient cultivars significantly increased in the upper and middle layers and significantly decreased in the lower layer as the amount of N applied increased.

**Table 6.** The vertical distribution of $P_n$, $T_r$, and $WUE_{inst}$ of two N-efficient and two N-inefficient winter wheat varieties under different N levels.

| | $P_n$ (μmol·m$^{-2}$·s$^{-1}$) | | | $T_r$ (g·m$^{-2}$·h$^{-1}$) | | | $WUE_{inst}$ | | |
|---|---|---|---|---|---|---|---|---|---|
| | UL | ML | LL | UL | ML | LL | UL | ML | LL |
| **Cultivar** | 5401.17 ** | 2366.23 ** | 12,255.6 ** | 7963.05 ** | 22.82 ** | 67.55 ** | 2196.24 ** | 19,895.15 ** | 139,761.08 ** |
| XN509 (TN-in) | 15.83d | 13.60d | 6.62c | 3.78d | 3.99d | 0.67d | 4.22c | 3.41b | 9.93a |
| AK58 (TN-in) | 19.28c | 16.38c | 9.21b | 4.68c | 4.62c | 1.85c | 4.21c | 3.56b | 5.35b |
| YM49 (TN) | 27.22b | 24.36b | 9.71a | 5.47a | 4.93b | 2.06b | 4.97b | 4.95a | 4.67c |
| ZM27 (TN) | 28.84a | 25.41a | 9.27b | 5.36b | 5.29a | 2.16a | 5.39a | 4.87a | 4.31d |
| **N level** | 2135.57 ** | 1115.76 ** | 2656.9 ** | 47.43 ** | 1.94 ** | 0.7 ** | 26,349.78 ** | 41,802.06 ** | 3344.43NS |
| N8 | 20.83b | 19.71b | 11.14a | 4.52b | 4.62b | 2.18a | 4.56b | 4.24a | 6.26a |
| N15 | 24.75a | 20.15a | 6.24b | 5.06a | 4.79a | 1.19b | 4.83a | 4.15b | 5.86b |
| **N level × Cultivar** | 170.39 ** | 104.95 ** | 6319.3 ** | 231.46 ** | 14 ** | 9.74 ** | 3133.45 ** | 8851.69 ** | 13,269.02 ** |

TN-in: The N-inefficient cultivars, TN: The N-efficient cultivars. Results of split–split plot variance analyses. NS, and ** indicate non-significant, significant at *p* ≤ 0.05, and significant at 0.01, respectively. Different letters within each column indicate significant differences according to Duncan's multiple-range test (*p* = 0.05). The main effects were compared using Student's *t*-test.

### 3.4. Vertical Distribution of the Leaf N Content and PNUE

#### 3.4.1. Vertical Distribution of the Leaf N Content

The leaf N content was 8.3–78%, 17.4–82.4%, and 4.9–46.7% higher in the N-efficient cultivars than in the N-inefficient cultivars in the upper, middle, and lower layers, respectively, and these differences were significant (Figure 6). The leaf N content of the N-efficient and N-inefficient cultivars was highest in the upper layer, followed by the middle and lower layers, and differences in the leaf N content among layers were significant. The canopy leaf N content of the N-efficient and N-inefficient cultivars increased significantly as the amount of N applied increased.

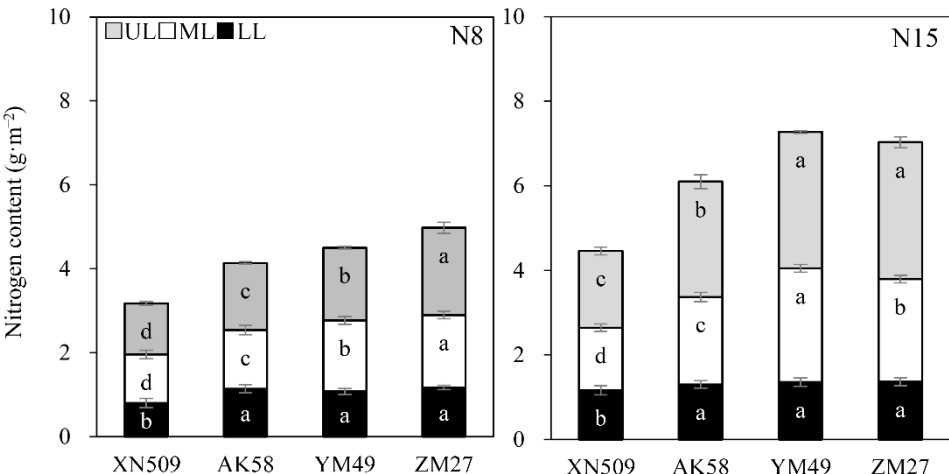

**Figure 6.** The vertical distribution of the leaf N content of two N-efficient and two N-inefficient winter wheat varieties under different N levels. UL: upper layer. ML: middle layer. LL: lower layer. Different letters within each column indicate significant differences according to Duncan's multiple-range test (*p* = 0.05).

### 3.4.2. Vertical Distribution of PNUE

The PNUE values were 8.1–48.27% and 12.6–25.0% higher in the N-efficient cultivars than in the N-inefficient cultivars in the upper and middle layers in the N8 treatment, respectively, and these differences were significant. In this same treatment, the PNUE values were only 19.75–25.23% higher in the N-efficient cultivars than in XN509 (these differences were significant); however, there were no significant differences in the PNUE of the N-efficient cultivars and AK58 (Figure 7). There were no significant differences in the PNUE of the N-efficient cultivars and XN509 in the upper layer in the N15 treatment; however, the PNUE was 12.1–14.2% higher in the N-efficient cultivars than in AK58, and these differences were significant. The PNUE was 3.8–47.8% and 2.3–60.4% higher in the N-efficient cultivars than in the N-inefficient cultivars in the middle and lower layers, respectively, and these differences were significant. The canopy PNUE of the N-efficient and N-inefficient cultivars increased significantly as the amount of N applied increased.

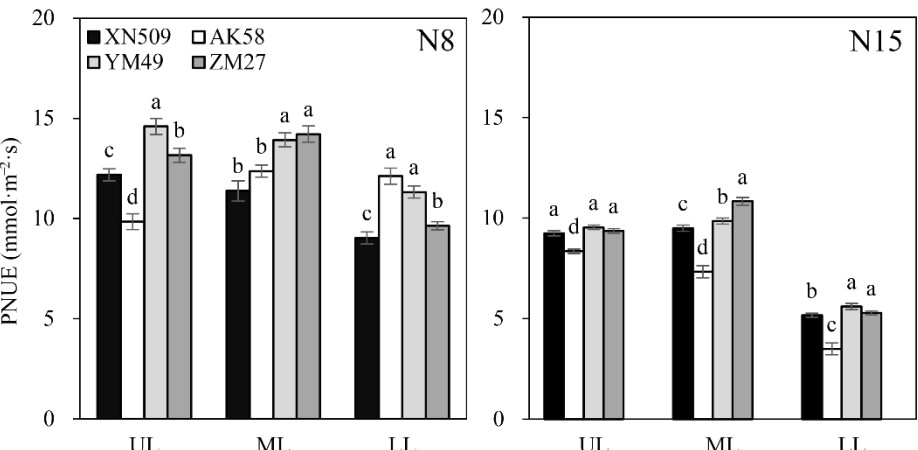

**Figure 7.** The vertical distribution of PNUE of two N-efficient and two N-inefficient winter wheat varieties under different N levels. UL: upper layer. ML: middle layer. LL: lower layer. Different letters within each column indicate significant differences according to Duncan's multiple-range test (*p* = 0.05).

3.4.3. $K_L$, $K_N$, and $K_N/K_L$

The $K_L$ values of the N-efficient cultivars YM49 and ZM27 were 0.362 and 0.339 in the N15 treatment, respectively. $K_L$ values were 31.6–34.6% lower in the N-efficient cultivars than in the N-inefficient cultivars in the two N treatments, and these differences were significant (Tables 7 and S5). These findings indicate that the N-efficient cultivars were more compact than the N-inefficient cultivars. The $K_N$ of the N-efficient cultivars YM49 and ZM27 were both 0.219 in the N15 treatment. The $K_N$ values were 31.7–52.27% higher in the N-efficient cultivars than in the N-inefficient cultivars in the two N treatments, and these differences were significant. These findings indicate that the leaf N content of the N-efficient cultivars was distributed in the upper and middle layers of the canopy to a greater degree compared with the leaf N distribution of the N-inefficient cultivars. The $K_N/K_L$ values of the N-efficient cultivars YM49 and ZM27 were 0.605 and 0.646 in the N15 treatment, respectively. The $K_N/K_L$ values were 78.4–103.8% higher in the N-efficient cultivars than in the N-inefficient cultivars in the two N treatments, and these differences were significant. These findings indicate that the distributions of light and N were more consistent in the canopy of the N-efficient cultivars than in the canopy of the N-inefficient cultivars. The $K_L$, $K_N$, and $K_N/K_L$ of the N-efficient and N-inefficient cultivars increased significantly as the amount of N applied increased.

**Table 7.** $K_L$, $K_N$, and $K_N/K_L$ of two N-efficient and two N-inefficient winter wheat varieties under different N levels.

|  | $K_L$ | $K_N$ | $K_N/K_L$ |
|---|---|---|---|
| **Cultivar** | 2075.93 ** | 1245.47 ** | 17,614.42 ** |
| XN509 (TN-in) | 0.454b | 0.132d | 0.290c |
| AK58 (TN-in) | 0.459a | 0.145c | 0.310c |
| YM49 (TN) | 0.345c | 0.191b | 0.553b |
| ZM27 (TN) | 0.341d | 0.201a | 0.591a |
| **N level** | 361.25 ** | 1208.15 ** | 2895.85 ** |
| N8 | 0.386b | 0.151b | 0.404b |
| N15 | 0.413a | 0.184a | 0.469a |
| **N level × Cultivar** | 503.59 ** | 556.32 ** | 942.34 ** |

TN-in: The N-inefficient cultivars, TN: The N-efficient cultivars. Results of split–split plot variance analyses. ** indicate significant at $p \leq 0.05$. Different letters within each column indicate significant differences according to Duncan's multiple-range test ($p = 0.05$). The main effects were compared using Student's *t*-test.

## 4. Discussion

### 4.1. Relationships among Leaf Morphological Traits, Plant Type, and the Canopy Light Distribution

The structure of the canopy is determined by the orientation and area of the leaves. The spatial distribution of plants and LAI are important indicators that reflect the ability of crops to intercept light within the crop canopy [40,41]. In our study, FLL, FLW, FLA, FLP, FLNW, and BA were significantly higher in the N-efficient cultivars than in the N-inefficient cultivar XN509 and significantly lower in the N-efficient cultivars than in the N-inefficient cultivar AK58. DA was significantly lower in the N-efficient cultivars than in the N-inefficient cultivars, and the opposite pattern was observed for LC. These differences indicated that the excessively large and flat leaves of the N-inefficient cultivars result in the draping of leaves; the higher LC and lower DA of the N-efficient cultivars support this finding. Less area of the leaves is exposed to light when leaves are small, such as in XN509. Previous studies have indicated that erect leaves that experience less pronounced draping as in the N-efficient cultivars might increase the compactness of wheat plants [42–44]. When the canopy leaves are closer to the stem as a whole, the sun shines deeper, which means that $K_L$ is going to be smaller [22]. In our study, we got similar results: the PAR was significantly higher in the N-efficient cultivars than in the N-inefficient cultivars in the middle layer, and the $K_L$ (YM49 = 0.345 and ZM27 = 0.341) in the N-efficient cultivars was significantly lower than that in the N-inefficient cultivars. The light distribution in the canopy of the N-efficient cultivars is relatively more uniform. Light is an important

factor in plant biomass accumulation [42]. In fact, many studies have pointed out that the canopy LAI and biomass of the N-efficient cultivars are always higher [45,46], as demonstrated in our study. Obviously, the smaller $K_L$ increases the distribution of light and enhanced biomass accumulation to the middle and lower layers. Overall, the compact plant type is the rational evolution of the N-efficient cultivars YM49 and ZM27 to optimize the canopy resources.

### 4.2. Relationship between the Canopy Light and N Distribution

Heterogeneities in canopy microclimates have a substantial effect on canopy photosynthesis [29,46]. In our study, the $P_n$, $T_r$, $WUE_{inst}$, N content, and PNUE were significantly higher in the N-efficient cultivars YM49 and ZM27 than in the N-inefficient cultivars XN509 and AK58 in the upper and middle layers. The supply of light energy associated with the light gradient in the canopy of the N-efficient cultivars can promote canopy photosynthesis [47]. Higher $P_n$ might reflect higher levels of photosynthesis and chlorophyll in the middle and upper layers, a stronger C fixation ability, and higher demand for C and N in the leaves [48]. Higher Tr and $WUE_{inst}$ might enhance the transport of biomass and nutrients, which promotes the metabolism of leaves, helps maintain curled and erect leaf shapes [16,45,49], and facilitates the flow of C and N in the shaded leaves and stems to the middle and upper layers of the canopy [1,50]. $K_N$ (YM49 = 0.191 and ZM27 = 0.201) was larger in the N-efficient cultivars than in the N-inefficient cultivars, which indicates that more N was distributed in the middle and upper layers in the N-efficient cultivars in our study. The N gradient can alleviate photoinhibition of the leaves at the top of the canopy under strong light conditions and facilitate the maintenance of high PNUE, thereby reducing C and N loss [51]. In N-inefficient cultivars, a lack of light reduces photosynthetic activity in the middle and lower layers and might result in the transport of N and C involved in photosynthesis to the grains or a yield loss of 20–50% in the form of photorespiration [40,51–53]. Therefore, the superior canopy structure of the N-efficient cultivars enhances the distribution of light and N in the canopy, and this might be the mechanism by which canopy structure controls the photosynthetic capacity of the canopy and NUE [54].

### 4.3. Relationship between the Canopy Light and N Distribution with Yield and NUE

The NUE of crops is mainly affected by genotype, cultivation method, and environmental conditions [55,56]. In our study, the NUtE, NUpE, and yield of the N-efficient cultivars YM49 and ZM27 were significantly higher compared with those of the N-inefficient cultivars XN509 and AK58. NUE significantly differs among genotypes and is highly correlated with yield [32,57]. Previous studies have shown that more than 90% of the yield of crops is derived from photosynthetic products during grain filling [58], and the accumulation of photosynthate during this period was limited by light and leaf nitrogen content [59]. The light distribution in the canopy affects the photosynthetic rate and distribution of N in canopy leaves [38]. There is a lot of strong evidence that the relatively more uniform light and N distribution of the N-efficient cultivars increased the 1000-grain weight [22,42,47] and in turn, the $K_N/K_L$ values, which indicate higher efficiency of canopy light and N utilization. In our study, the 1000-grain and $K_N/K_L$ (YM49 = 0.553 and ZM27 = 0.591) of N-efficient cultivars YM49 and ZM27 were significantly higher compared with that of the N-inefficient cultivars XN509 and AK58, as found in other studies. This might explain why the aboveground dry matter, light energy utilization, and NUE were significantly higher in the N-efficient cultivars than in the N-inefficient cultivars [22,60].

In general, the most ideal goal is to build a "smart canopy", in which the distribution of light and nitrogen can automatically and dynamically adjust, and promote the coordination between wheat individuals, rather than competition [61]. In our study, LC, DA, $K_L$, $K_N$, and $K_N/K_L$ can be used as potential reference indicators for smart canopies. In order to deepen our understanding of the physiological and biochemical processes of wheat population light and nitrogen, the following subjects deserve further research: the impact of a "smart canopy" on the light environment of a wheat field canopy, and the improvement of the

efficiency of light interception and nutrient assimilation through genetic control through breeding or genetic engineering/editing strategies in the whole canopy to improve yield, especially in the field fluctuating environment.

## 5. Conclusions

Differences in canopy structure between N-efficient and N-inefficient winter wheat cultivars lead to changes in the light and N distribution and photosynthetic capacity, which affect light energy-use efficiency, NUE, 1000-grain weight, and yield. The N-efficient cultivars YM49 and ZM27 had more moderate flag leaf sizes and less pronounced leaf draping and curling compared with the N-inefficient cultivars. The low $K_L$ and compactness of the N-efficient cultivars enhanced the light conditions and increased the allocation of light to the middle and lower layers of the canopy. The ventilated and transparent population had higher $P_n$, $T_r$, $WUE_{inst}$, and PNUE and lower N remobilization of the upper and middle canopy leaves, which increased the specific leaf N content. Improved canopy structure and effective distribution of light and N are associated with higher $K_N/K_L$ values, which reflect the assimilation ability of N-efficient winter wheat cultivars, enhance 1000-grain weight and grain yield, and significantly improve light and NUE. The results of our study indicate that a "smart canopy" involves a better match between light and N dynamics in the canopy, which results in an improved balance between C and N metabolism and reduces the loss of energy and nutrients. Achieving this "smart canopy" should be the goal of future research aimed at breeding N-efficient cultivars.

**Supplementary Materials:** The following supporting information can be downloaded at: https://www.mdpi.com/article/10.3390/agronomy12102411/s1. Table S1. Yield, yield components, and NUtE of two N-efficient and two N-inefficient winter wheat varieties under different N levels. Table S2. Flag leaf morphological traits of two N-efficient and two N-inefficient winter wheat varieties under different N levels. Table S3. The leaf angle vertical distribution of two N-efficient and two N-inefficient winter wheat varieties under different N levels. Table S4. $P_n$, $T_r$, and $WUE_{inst}$ vertical distribution of two N-efficient and two N-inefficient winter wheat varieties under different N levels. Table S5. $K_L$, $K_N$, and $K_N/K_L$ of two N-efficient and two N-inefficient winter wheat varieties under different N levels.

**Author Contributions:** Conceptualization, Z.Z. and S.X. (Shuping Xiong); Data curation, S.X. (Saijun Xu); Formal analysis, Q.W. and Y.Y.; Funding acquisition, X.M. and S.X. (Shuping Xiong); Investigation, H.P., X.F., and Z.F.; Methodology, B.Q.; Project administration, X.M. and X.W.; Resources, S.X. (Shuping Xiong); Software, Q.W.; Supervision, S.X. (Shuping Xiong); Visualization, Z.Z.; Writing original draft, Z.Z. and S.X. (Saijun Xu); Writing review and editing, S.X. (Shuping Xiong). All authors have read and agreed to the published version of the manuscript.

**Funding:** This research was funded by "National Key Research and Development Program of China (2021YFD1700904)", "Key Research Projects of Henan Higher Education Institutions (21A210015)," "Key R & D and promotion projects in Henan Province (212102110048)", "National Natural Science Funds of China (32071956)" and "Major science and technology project of Henan Province (221100110800)".

**Data Availability Statement:** Not applicable.

**Acknowledgments:** We thank Gege Zong for providing suggestions that greatly improved the manuscript.

**Conflicts of Interest:** The authors declare no conflict of interest.

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
