# Peer review of "Variation in Leaf Type, Canopy Architecture, and Light and Nitrogen Distribution Characteristics of Two Winter Wheat (Triticum aestivum L.) Varieties with High Nitrogen-Use Efficiency"

_agronomy, doi:10.3390/agronomy12102411_

Round 1
Reviewer 1 Report
Overall I think this is a nice paper and of interest but my one main concern is that it is very dense and difficult to read throughout. I got confused by the different varieties and all the different stats and values. I also think the results need to be removed from the discussion and it focused more on putting your work in context.
Abstract reads nicely and is comprehensive, my one comment is that it is quite dense to read with the list of all features that are affected. It could be worth using slightly shorted sentences for ease of reading.- same in the intro for example line 43 onwards
Ensure the L is subscript in KL in the abstract to match latter text
Line 49- developed countries like where? A couple of examples would help put this in context
Being very picky and I know the paper is about N but it is written almost every line
Table 1- consider adding to the legend that these values were before any fertilisers were added
Line 143- why were par values taken between 2 and 4 and not around solar noon?
Line 158- what conditions did you use inside the licor chamber? i.e. PAR, leaf temp, CO2, humidity etc. or did you have an open ‘sun and sky’ chamber?
Line 197- is this sentence about ethical approval meant to be here..?
For table 2 it could be worth adding which are the cultivars are N efficient and which are N inefficient
Lines beginning 221- is very difficult to read with all the % values and mention of the different cultivars. It could be worth sticking to the main differences in the text and have the % differences in the table or a graph to make it easier to see. The same goes for the rest of the results section- it is so dense it is hard to see the important information.
The start of the discussion is good putting the results in context and tying them together but towards line 400, it starts to just repeat the results section.
Section 4.2 puts your results in context of others well but section 4.3 again goes back to repetition of results. Try to stop this and focus on the key features and what it means. Line 436 onwards seems more like introductory information as opposed to discussion unless you tie it to your results and those of others.
I expected to see a paragraph in the discussion talking about the smart canopy or the design of a smart canopy with improved NUE and light interception using the results you have found here. And which of your efficient may be most suitable/ where you would expect changes etc.
Author Response
Comments and Suggestions for Authors
1.Overall I think this is a nice paper and of interest but my one main concern is that it is very dense and difficult to read throughout. I got confused by the different varieties and all the different stats and values. I also think the results need to be removed from the discussion and it focused more on putting your work in context.
Thanks to the reviewers for your evaluation of this article, some of the results in the article do look a little difficult to read due to the relationship between varieties and nitrogen treatment. In response to the comments and to make the paper easier to read, we have removed some of the stats and values form the discussion and make the discussion more logical and focused. See the modified part of the discussion for details.
2.Abstract reads nicely and is comprehensive, my one comment is that it is quite dense to read with the list of all features that are affected. It could be worth using slightly shorted sentences for ease of reading.- same in the intro for example line 43 onwards
We are grateful to the reviewer’s suggestions and comments. In response to the comments and to make the language more concise and easier to read, we have simplified the long sentences in this paper. See the modified part of line 34,45,54,59,80 for details.
3.Ensure the L is subscript in KL in the abstract to match latter text
We’re very sorry for the formatting errors. We have modified it in the abstract to match later text, which rectified at line 23 and 27. And we have also checked the KL format in the whole text.
4.Line 49- developed countries like where? A couple of examples would help put this in context
We’ve added examples to line 50 and provide relevant references. Actually, China’s NUE was lower than that in North America or Europe where it was typically above 50%,
5.Being very picky and I know the paper is about N but it is written almost every line
We’re sorry to bother you, but the nitrogen is the center of our research, and it is so important that it appears frequently in papers. We have tried to avoid N from appearing in each line in the revised draft.
6.Table 1- consider adding to the legend that these values were before any fertilizers were added
Thank you for the reviewer’s consideration, perhaps we did not make it clear in our writing. Table-1 indicated the soil base fertility before nitrogen treatment, it is rectified at Line 99 and 114.
7.Line 143- why were par values taken between 2 and 4 and not around solar noon?
Thanks for the reviewer’s questions, we may not have explained it clearly in the text. When AccPAR LP-80 measuring in the field, it needs to be connected to an external sensor to collect data from the upper part of the wheat canopy. That measurement ranges from 0 to 2000 μmol M-2s-1. Because the light radiation intensity usually exceeds this range at solar noon, we decided not to use at solar noon.
8.Line 158- what conditions did you use inside the licor chamber? i.e. PAR, leaf temp, CO2, humidity etc. or did you have an open ‘sun and sky’ chamber?
Thanks for the reviewer’s questions, we are measured in an open ‘sun and sky’ chamber,and we add the explanation at line 163 (unmarked mode).
9.Line 197- is this sentence about ethical approval meant to be here...?
We are very sorry for the mistakes for using the writing template given by the journal, and it had been already deleted.
10.For table 2 it could be worth adding which are the cultivars are N efficient and which are N inefficient
We have added it to table-2 and other tables.
11.Lines beginning 221- is very difficult to read with all the % values and mention of the different cultivars. It could be worth sticking to the main differences in the text and have the % differences in the table or a graph to make it easier to see. The same goes for the rest of the results section- it is so dense it is hard to see the important information.
We are grateful to the reviewer’s suggestions and comments. In response to the comments and to make the language more concise and easier to see, we have rewritten the section of 3.2.1. And some of the rest of the result section has also been modified.
12.The start of the discussion is good putting the results in context and tying them together but towards line 400, it starts to just repeat the results section.
We have reorganized the statements here and added further discussion and conclusion.
Section 4.2 puts your results in context of others well but section 4.3 again goes back to repetition of results. Try to stop this and focus on the key features and what it means. Line 436 onwards seems more like introductory information as opposed to discussion unless you tie it to your results and those of others.
We have reorganized the statements here and added further discussion and conclusion. The section of 4.3 has been simplified to remove some statements and added discussion.
13.I expected to see a paragraph in the discussion talking about the smart canopy or the design of a smart canopy with improved NUE and light interception using the results you have found here. And which of your efficient may be most suitable/ where you would expect changes etc.
We are grateful to the reviewer’s suggestions, we have added a paragraph in the discussion of section 4.3. See the modified part of line 452~461 for details (unmarked mode).

Reviewer 2 Report
Dear appreciated Authors,
The presented paper should be accepted for publication, after minor revisions because the paper represents a great contribution to science. However to improve the quality of paper I suggest the following:
Line 43-47: I suggest to divide sentence on two, as a: The N remaining after it is utilized by crops to support their growth is rapidly lost to the environment through ammonia volatilization, denitrification, surface runoff and other pathways [6]. This results in soil consolidation, reductions in crop yield and quality [7], eutrophication and acidification of aquatic and terrestrial ecosystems [8], and the formation of nitrous oxide, which contributes to the greenhouse effect [9,10].
Line 53-55: For this reason, enhancing the NUE of winter wheat is important for achieving food security and preventing environmental degradation both in China specifically, as well as in the world.
Line 214: Grain number per spike (together in one row, should not divide in the bottom row since that - per spike is not unit of measurement . If there is not enough place for whole parameter name in the table it could be written in abbreviations and then explain abbreviations below the table below the table.
Example: SP: Spike number, GNS: Grain number per spike, GY: Grain yield.
Best regards,
NL

Author Response
Comments and Suggestions for Authors
Dear appreciated Authors,
1.The presented paper should be accepted for publication, after minor revisions because the paper represents a great contribution to science. However to improve the quality of paper I suggest the following:
Line 43-47: I suggest to divide sentence on two, as a: The N remaining after it is utilized by crops to support their growth is rapidly lost to the environment through ammonia volatilization, denitrification, surface runoff and other pathways [6]. This results in soil consolidation, reductions in crop yield and quality [7], eutrophication and acidification of aquatic and terrestrial ecosystems [8], and the formation of nitrous oxide, which contributes to the greenhouse effect [9,10].
Line 53-55: For this reason, enhancing the NUE of winter wheat is important for achieving food security and preventing environmental degradation both in China specifically, as well as in the world.
We appreciate for your suggestions, we have simplified the long sentences in this paper. And see the modified part of line 43~47, 53~55 for details (unmarked mode).
2.Line 214: Grain number per spike (together in one row, should not divide in the bottom row since that - per spike is not unit of measurement . If there is not enough place for whole parameter name in the table it could be written in abbreviations and then explain abbreviations below the table below the table.
Example: SP: Spike number, GNS: Grain number per spike, GY: Grain yield.
We are grateful to the reviewer’s suggestions, we agree with you that consider adding to the legend that these values were before any fertilizers were added, it is rectified at Table-2.
